# Rheological Method for Determining the Molecular Weight of Collagen Gels by Using a Machine Learning Technique

**DOI:** 10.3390/polym14173683

**Published:** 2022-09-05

**Authors:** Karina C. Núñez Carrero, Cristian Velasco-Merino, María Asensio, Julia Guerrero, Juan Carlos Merino

**Affiliations:** 1Department of Condensed Matter Physics, University of Valladolid, 47011 Valladolid, Spain; 2Foundation for Research and Development in Transport and Energy (CIDAUT), 47051 Valladolid, Spain

**Keywords:** collagen analysis, rheology, collagen structure and modelling, molecular weight, machine learning tools

## Abstract

This article presents, for the first time, the results of applying the rheological technique to measure the molecular weights (Mw) and their distributions (MwD) of highly hierarchical biomolecules, such as non-hydrolyzed collagen gels. Due to the high viscosity of the studied gels, the effect of the concentrations on the rheological tests was investigated. In addition, because these materials are highly sensitive to denaturation and degradation under mechanical stress and temperatures close to 40 °C, when frequency sweeps were applied, a mathematical adjustment of the data by machine learning techniques (artificial intelligence tools) was designed and implemented. Using the proposed method, collagen fibers of Mw close to 600 kDa were identified. To validate the proposed method, lower Mw species were obtained and characterized by both the proposed rheological method and traditional measurement techniques, such as chromatography and electrophoresis. The results of the tests confirmed the validity of the proposed method. It is a simple technique for obtaining more microstructural information on these biomolecules and, in turn, facilitating the design of new structural biomaterials with greater added value.

## 1. Introduction

It is well known that biomacromolecules, such as collagen from many species, including bovine, porcine, and fish,, provide an option for the fabrication of advanced bio-based materials [1]. Several previous authors found potential applications of collagen in industries as diverse as the pharmaceutical and biomedical industries, as collagen provides improves physical–chemical properties, mechanical stability, strength, biocompatibility, and biodegradability [2,3,4,5]. Collagen has potential applications in the cosmetics industry because of its properties as natural humectant and moisturizer for the skin [6]. Collagen has been used in food products because it improve the elasticity, consistency, and stability of those products; in addition, collagen improves the quality, nutritional value, and health value of food products [7,8,9]. Collagen has even been used in applications as novel as the molding of thermoplastic collagen parts by additive manufacturing [10].

Most of the collagen varieties used in these applications resulted from applying thermomechanical and chemical procedures to native collagen to obtain a hydrolyzed species and/or molecular weights that were significantly lower than those of native collagen [11]. Numerous previous authors described the successful use of traditional characterization techniques to measure the Mw of these types of structures [12]. However, to explore new applications in which maintaining a high Mw is important (e.g., high mechanical performance applications), existing characterization techniques may have limitations in assessing these high Mw collagen fibers. These limitations pose a problem when designing collagen fiber-based biomaterials for such applications, as the structure-property relationship will be difficult to establish.

The main problem is that the triple helix of collagen fibers is highly sensitive to degradation/denaturation in the presence of solvents or high temperatures [13]. Traditional Mw measurement assays require the preparation of solutions that can affect the structure. For example, techniques, such as sodium dodecyl sulfate-polyacrylamide gel electrophoresis (SDS-PAGE), require the use of dilutions to make qualitative measurements of Mw. Additionally, SDS-PAGE is a technique that cannot measure MwD or the polydispersity index (PDI). Shi et al. [14] studied the polymorphism and stability of the nanostructures of three types of collagen by SDS-PAGE: bovine flexor tendon, rat tail, and tilapia skin. The technique allowed the differences in Mw between the three collagen types to be qualitatively distinguished (all samples were subjected to the same digestion). The maximum measured molecular weight depended on the reference used, demonstrating that SDS-PAGE is a qualitative technique. Xu et al. [15], Bhuimbar et al. [16], and Zhang et al. [17]; applied the SDS-PAGE method successfully, but for low-molecular-weight collagen samples.

In addition to SDS-PAGE, mass spectrometry by matrix-assisted laser desorption/ionization with a time-of-flight type ion detector (MALDI-TOF) is traditionally a highly sensitive technique, but is limited to measurements of low Mw. For this reason, its use is widely reported in studies in which low-molecular-weight collagen is used. For example, Kim et al. [18] and Kumar et al. [19] reported the results of chemically grafting different types of low-Mw collagen. They used MALDI-TOF to examine the microstructure. Their results demonstrated that the MALDI-TOF technique was suitable for monitoring the success of the graft under study, but not for exactly measuring molecular weights and their distributions.

Another traditional method is the size-exclusion chromatography (SEC) technique. Although it is the most powerful traditional technique, it is the most complex from an experimental point of view, as it requires several prior optimizations, such as sample dilution, column selection, and selection of the mobile phase that serves as a vehicle through the measurement column [18]. A review of the existing literature on the use of the SEC technique to measure collagen Mw showed that there are few examples of such use, especially in recent years. This is mainly due to the complexity of the SEC technique and the fact that sample preparation involves dissolution in all cases, which means that only solubilized fibers can be measured. There are references in the literature to successful studies of Mw and its distribution in low Mw collagen, such as fish collagen, gelatines, or hydrolysates, including those of Cozza et al. [20] and Meyer et al. [21].

In contrast, the use of the rheometric technique to measure Mw has been widely reported for conventional synthetic macromolecules. This method is based on translating dynamic rheological measurements into quantitative Mw values and involves an understanding of the macromolecular relaxations and transitions of polymeric chains. The available knowledge about the rheometric technique is derived from the pioneering works of Wu [22] and Tuminello [23,24]. All of the equations proposed for the calculation of Mw distribution from simple melt rheology tests have been validated by multiple authors on different materials [25,26]. Software has been developed to automatically translate all rheological information into Mw information for a variety of synthetic polymers [27,28]. The rheometric technique has gained relevance in the field of polymers because it dispenses with the use of solvents and complex sample preparations. Other advantages include its low costs and the ease of interpreting its results.

Zhang et al. [29] pioneered the application of the rheological technique in measuring Mw in biomacromolecules, such as cellulose. Their results demonstrated the potential of the technique by validating it with conventional gel permeation chromatography (GPC) assays. However, the use of the rheological technique on collagen materials has not been explored to date.

Lai et al. [30] carried out studies on the dynamic viscoelastic properties of collagen solutions, which was a good starting point for translating the rheological behavior of collagen samples into Mw, using existing software. However, those studies were performed on solubilized collagen samples. Therefore, there was a challenge in finding the right test conditions to obtain rheological information from long, non-soluble collagen fibers. In recent years, the literature confirms the rise of artificial intelligence techniques to address similar challenges in the identification and characterization of biomolecules [28,29,30] and particle forms in predicting the Mw of complex systems [31].

The present study aims to present, for the first time, the results of applying the rheological technique to measure Mw of non-soluble collagen fibers. Our study was based on rheological data obtained under conditions of maximum molecular mobility during frequency sweeps. This required the use of artificial intelligence techniques to fit the experimental data.

## 2. Materials and Methods

### 2.1. Materials

Fresh bovine skin was skinned and neutralized by a process described by Zhang et al. [32]. A solution of 96% acetic acid from Sigma-Aldrich, Spain (0.5 M) was used to obtain the collagen dispersions that were necessary for the rheological study.

### 2.2. Methods

#### 2.2.1. Collagen Sample Dilution Process

Diluted samples of collagen were prepared in acetic acid at concentrations of 1, 2, 3, 4, 5 and 15% w/w, using a Haake Rheomix internal mixer (ThermoFisher Scientific SL, Germany) at 4 °C and 25 rpm for 2 h. All insoluble gels with a pH value of 2.5 were centrifuged for 15 min at 4 °C to remove trapped air bubbles. The samples were stored for no longer than 48 h before assaying. It is important to note that the prepared samples retained both soluble and non-soluble fractions for the study of all macromolecular sizes.

#### 2.2.2. Obtaining Collagen Samples of Different Molecular Weights

In addition to the original samples obtained at different concentrations, samples of different molecular weights were used to validate the method. To obtain samples of different Mw, the original collagen mass was exposed to a thermomechanical treatment in a Haake Rheomix internal mixer (30 °C, 35 rpm, with roller-type mixers) at different times (5, 10, and 15 min). Using this method, breaks in their chains were randomly induced. The resulting masses were diluted to 4% w/w using the procedure described above. These samples were labelled 4% wt_5, 4% wt_10, and 4% wt_15, respectively. The flow index was determined by measuring the viscosity at 32 °C and 2.16 kg. The tests determined that the viscosity of the samples decreased with treatment time 4% wt_5 > 4% wt_10 > 4% wt_15, so the Mw was expected to follow this trend.

#### 2.2.3. Rheological Tests

Oscillatory rheological experiments were performed on an oscillatory shear in a Haake RS-600 Rheometer, using stainless steel cone/plate geometry with a gap set between 1 mm and 2 mm, depending on the sample. Dynamic rheological properties (storage modulus G′ and loss modulus G″) were obtained in the linear and non-linear viscoelastic region at temperatures from 23 °C to 40 °C with an accuracy of ± 0.1 °C. The linear viscoelastic region was denoted where the values of the modulus were independent of stress or strain. The data were analyzed with RheoWin software V.4.86.0002 (Thermo Fisher Scientific, GermanyDynamic frequency sweeps were performed for collagen gels, with different concentrations in different test modes. The test conditions are set out in Table 1. Multiple combinations of experimental parameters (inside and outside the viscoelastic range) were studied to find the cut-off points of G′ and G″.

#### 2.2.4. Application of Machine Learning Techniques

Open source Python 3. x toolkits V.3.9., such as Numpy, Pandas, Matplotlib, scipy.optimize.curve_fit (Scipy), and Sklearn, were used to perform the automatic model fitting and the generation of theoretical values. This was necessary due to the high sensitivity of the samples to mechanical stress and temperature during the rheological tests. The fitted curves were fed into Rheowin [28] software for the construction of the Mw Gaussians. The mathematical principle of the calculations performed by the software is explained in Section 3.

#### 2.2.5. Statistical Analysis

The oscillatory tests were repeated a minimum of three times per sample; the standard deviation of the measurements was reported. Experimental data sets, from the different data populations that were obtained, were analyzed by analysis of variance (ANOVA) at a 95% confidence level, using Minitab^®^ 17.0 software.

#### 2.2.6. Mw Measurements by Traditional Techniques

The lower molecular weight sample (4% wt_15) was tested by gel permeation chromatography (GPC) and sodium dodecyl sulfate-polyacrylamide gel electrophoresis (SDS-PAGE), to compare the sample with the data obtained from the proposed rheological method.

*GPC procedure:* For the determination of the molecular weight of the samples by GPC analysis, an Agilent 1260 HPLC apparatus consisting of a quaternary pump (G1311B), an automatic injector (G1329B), a thermostated column compartment (G1316A), a refractive index (IR) detector (G1362A), and a double-angle static light scattering (DALS) detector (G7800A) was used. Four columns were used for chromatographic separation: Poteema pre-column (5 μm, 8, 50 mm), Proteema 1000 Å (5 μm, 8, 300), Proteema 300 Å (5 μm, 8, 300), and Proteema 100 Å (5 μm, 8, 300). The freeze-dried collagen samples were dissolved in the mobile phase (0.15 M sodium acetate, 0.2 M acetic acid, pH 4.5) (1 mg/mL) and kept under stirring for 72 h to ensure a better degree of dissolution. Subsequently, they were incubated at 100 °C for 20 min and filtered through a 0.2 μm polytetrafluoroethylene (PTFE) membrane filter. One hundred μL of the filtered solution was injected and chromatographed at a flow rate of 0.5 mL/min using an isocratic elution profile. The column temperature was maintained at 20 °C, the IR detector at 35 °C, and DALS at 30 °C. The DALS detector was calibrated with a polyethylene oxide (PEO) standard (PSS, Germany) of 10^6^ kDa and a polydispersity index of 1.05. The refractive index increments (dn/dc) were extracted from the work of Nomura et al. [33]. Data were analyzed using Agilent GPC/SEC A.02.01 software.

*SDS-PAGE procedure:* The freeze-dried collagen was suitably dissolved in 0.1 N acetic acid at a concentration of 4 mg/mL to obtain a dense, translucent solution. An aliquot of this solution was mixed with sample buffer in a 1:1 ratio and heated at 100 °C for 5 min. An aliquot (10 μL) of this mixture was loaded onto polyacrylamide gels (7% acrylamide and 0.24% bisacrylamide), size 100 mm × 750 mm × 0.75 mm, and prepared according to the methodology described by Laemmli [34]. Electrophoresis was performed using a Mini-Protean II Cell by Bio-Rad, US. The composition of the electrophoresis sample buffer was 10.52% glycerol, 21% sodium dodecyl sulphate (SDS), 0.63% sodium dodecyl sulphate (SDS) (7%), 0.63% dithiothreitol (DTT), and 0.5 M Tris-HCL (pH 6.8).

## 3. Results and Discussion

### 3.1. Mathematical Principle of the Rheological Method Used

As described in the manuals for the software used [28], the first step in the transformation from rheological properties to a mathematical function that relates molecular motions to chain size (*w*(*M*)) is the calculation of the linear relaxation spectrum, *H*(*τ*). This function can be appreciated from its relationship to the linear relaxation modulus *G*(*t*) [35]. The relaxation modulus *G*(*t*) of the generalized Maxwell model is provided by Equation (1):(1)$$G\left(t\right)=\sum_{i=1}^{N}G_{i}e^{\left(-\frac{t}{\lambda_{i}}\right)}\;[25,28]$$
and the storage and loss moduli G′ and G″ are provided by the Equations (2) and (3):(2)$$G^{\prime }\left(\omega \right)=\sum_{i=1}^{N}\frac{G_{i}{\left(\omega \lambda_{i}\right)}^{2}}{1+{\left(\omega \lambda_{i}\right)}^{2}}\;[25,28]$$
(3)$$G^{\prime \prime }\left(\omega \right)=\sum_{i=1}^{N}\frac{G_{i}{\left(\omega \lambda_{i}\right)}^{}}{1+{\left(\omega \lambda_{i}\right)}^{2}}\;[25,28]$$

Here, *G_i_* and *λ_i_* are the moduli and relaxation times, respectively, of the individual Maxwell elements.

The continuous relaxation spectrum is defined by letting the number of elements in the generalized Maxwell model increase to infinity so that *G*(*t*) can be represented in terms of continuous functions, *F*(*λ*), such that *F∙dλ* is the contribution to the modulus from relaxation times between *λ* and *dλ*. The relation between the relaxation modulus *G*(*t*) and the spectrum is provided by Equation (4):(4)$$G\left(t\right)=\int_{0}^{\infty }F\left(\lambda \right)e^{\left(-\frac{t}{\lambda }\right)}d\lambda \;[25,28]$$

Normally, the natural logarithm of the relaxation time, *λ*, i.e., *d.ln*(*λ*), is used for the spectrum, and the continuous spectrum, *H*(*λ*), is used in place of *F*(*λ*), where *H* = *F*∙*λ* and *H*∙*dln*(*λ*) = *F*∙*dλ*. The relationship between the relaxation modulus and *H*(*λ*) is provided by Equation (5):(5)$$G\left(t\right)=\int_{0}^{\infty }H\left(\lambda \right)e^{\left(-\frac{t}{\lambda }\right)}d\left(ln\lambda \right)\;[25,28]$$

The storage and loss moduli G′ and G″ are related to *H*(*λ*), as follows:(6)$$G^{\prime \prime }\left(\omega \right)=\int_{0}^{\infty }\frac{H{\left(\omega \lambda \right)}^{2}}{1+{\left(\omega \lambda \right)}^{2}}d\left(ln\lambda \right)\;[25,28]$$
(7)$$G^{\prime \prime }\left(\omega \right)=\int_{0}^{\infty }\frac{H{\left(\omega \lambda \right)}^{}}{1+{\left(\omega \lambda \right)}^{2}}d\left(ln\lambda \right)\;[25,28]$$

To effect the transformation of H(t) into w(M), an approximation formula based on the double-reptation rule is used. The (generalized) mixing rule is provided in Equation 8:(8)$$G_{r}\left(t\right)=G_{N}{\left(\int_{Me}^{\infty }F{\left(M,t\right)}^{1/\beta }w\left(M\right)\frac{dM}{M}\right)}^{\beta }\;[25,28]$$
where *Gr* is the reptation modulus, *G_N_* is the plateau modulus, and *M_e_* = *M_c_/2* is the entanglement molecular weight (*M_c_* is the critical molecular weight). *F*(*M*, *t*) denotes the relaxation kernel function, which describes the relaxation behavior of a molecular weight fraction with a molecular weight of M, and b is a parameter characterizing the mixing behavior. Several forms of relaxation kernel have appeared in the scientific literature. Maier et al. [36] made an evaluation. The one used by rheology software decays in an essentially exponential manner. The subscript “r” of the stress relaxation *G*(*t*) indicates that only the contributions of the reptation dynamics of the whole polymer chain are considered; the dynamics of the chain segments (Rouse modes), which only weakly depend on *w*(*M*), are not considered.

To obtain the relaxation spectrum from the rheological data *G′* and *G″,* and then to relate it to the Mw using Equation (8), two initial experiments were necessary to determine the definition of the linear viscoelastic range and the construction of the master curve. To determine the linear viscoelastic region of the sample, deformation sweeps and/or stress sweeps were performed at two extreme temperatures (between 23 °C and 40 °C). Knowing the limits of the linear viscoelastic region allowed us to determine the test range, where the sample did not suffer partial destruction of its structure. Figure 1 shows examples of the test performed on a 2% wt specimen. The tests were carried out for all of the samples studied. With this information, the experimental parameters for the rheological tests, shown in Table 1, were selected.

To ensure that there was no denaturation of the protein during the rheological tests, a temperature sweep was performed on the samples, setting a test frequency within the linear viscoelastic region. In Figure 2, an example of the estimation of the T_d_ (denaturation temperature) is presented. For the temperature sweep, the viscosity (red curve) and the tanδ (black curve), which is the quotient of the loss modulus between the storage modulus (*G″/G*′), were plotted. As shown in Figure 2, there were two ways to establish the T_d_ of the samples. The first was to identify at which temperature the viscosity decreased by about 50% of its initial value; the second was to identify at which temperature the tanδ value reached a maximum. Table 2 shows the T_d_ for all of the samples studied.

Once the denaturation temperature (maximum test temperature) and the linear viscoelastic region were obtained, frequency sweeps were performed at different temperatures (between 23 °C and 40 °C, depending on the sample) to obtain the values of the dynamic moduli that were needed for the calculation. At this stage, it was important to define the crossover frequency *G*′ = *G*″ (related to elastic and viscous behavior, respectively), as this point denoted the end of flow behavior, which was dominated mainly by molecular weight, and the beginning of rubbery behavior, where the effect of molecular entanglements predominated. It was also important to corroborate that with increasing test temperature the crossover where *G″ = G′* was at higher frequencies, because the increased temperature enabled the material molecules to be more mobile and to relax in less time.

#### 3.1.1. Application of Machine Learning Techniques

Because many rheological curves (frequency sweeps to obtain G′ and G″) showed inhomogeneity (random noise, no cut-off point G′ = G″, multiple cut-off points, or no increase in the cut-off frequency with temperature) in the collected measurements, either due to the composition of the samples (very low concentrations), tests were performed close to denaturation temperatures to favor molecular mobility or in conditions outside the elastic range, Artificial intelligence techniques had to be used to fit the data, smooth the curves, and extrapolate the expected cut-off points. The program was able to detect the last correct values and fit the correct values to a better curve/function to predict the “subsequent incorrect values”.

#### 3.1.2. Master Curve, Relaxation Spectrum, and Mw Gaussians Computation

From the experimental and theoretical (fitted) curves, the computer tool was used to construct the master curve, the relaxation spectrum, and the molecular weight distribution. The master curve provided insight into the viscoelastic behavior of the material over several decades of time at a fixed temperature. To construct the master curve, the time–temperature superposition principle was applied.

Once the master curve was obtained, the tool enabled the calculation of the relaxation time spectrum. A spectrum was generated because it was understood that the molecule chains of the material have different sizes and therefore each one has a characteristic relaxation time. Finally, from the relaxation time spectrum, the tool enabled the plotting of the molecular weight distribution by solving Equation (8). Figure 3 graphically summarizes an example (the 5% wt sample) of the steps followed by the software and the adjustment applied to the frequency sweeps at 34 °C to obtain the molecular weight Gaussians.

### 3.2. Effect of Collagen Concentration on the Rheology-Based Results

The procedure described above was applied to all of the samples at different concentrations. In most cases, when working at low concentrations (i.e., 1% wt) within the linear viscoelastic range, the curves described a gel-like behavior, i.e., there was a parallel between G′ and G″, and G′ was larger than G″ [37]. The example in Figure 4 shows this behavior. Even when the theoretical fit was applied, it was not possible to obtain the cut-off point for the 1% wt. sample at 23 °C and 28 °C. As the temperature increased, the curves improved markedly; however, from 30 °C, the sample slid off the plate during the test, and a separation of the phases (collagen fibers and the medium in which the sample was immersed or swollen) was observed.

Different test conditions were studied (i.e., different gaps, stresses, and strains were applied in the oscillatory test) and data outside the linear viscoelastic range could only be obtained for the lower concentration samples (1% wt to 2% wt). The Mw data shown in Table 3 were inconsistent (high standard deviation) for these concentrations, regardless of whether the fit was applied, due to the conditions under which the data were obtained for these samples.

As the concentration of the studied samples was increased, the linear viscoelastic range was broader and a larger gap of experimental conditions, within the linear viscoelastic regime, could be used for the oscillatory tests. However, as shown in Table 2, as the concentration of the samples increased, the motion constraints decreased the T_d_, so that (as shown in Table 3) the reported molecular weights were more reproducible and consistent only at low temperatures, within the linear viscoelastic range and when data fitting was applied, for samples between 3% wt and 5% wt.

Slightly increasing the concentration of the samples and finding the optimal conditions for the oscillatory tests enabled a wider molecular weight distribution to be obtained, i.e., more chains of different sizes could be detected, which increased the polydispersity index. These phenomena were more easily observed when the data were adjusted with the computer tool. This considerably improved the applicability of the method.

By moving the test temperatures away from Td, better rheological curves were obtained at concentrations close to 4% wt to 5% wt, in terms of dispersion and data fit (see Figure 5). These results were more homogeneous and indicated temperature-independent Mw values (see Figure 6 and Table 3). To statistically evaluate the experimental conditions under which the method is valid, an analysis of variance (ANOVA) was performed for different data populations (see Table 4). This analysis tested the hypothesis that the population means were equal (e.g., at different concentrations or different temperatures). ANOVAs assess the significance of one or more factors by comparing the means of the response variables at different factor levels. The null hypothesis states that all population means (factor level means) are equal (*p*-value > 0.05), while the alternative hypothesis states that at least one is different (*p*-value < 0.05) [38].

Exorbitant *p*-values were obtained at low concentrations, regardless of the test conditions or the fit applied to the data (runs 1 to 5; see Table 4). This demonstrated the poor feasibility and reproducibility of the method under these conditions. Furthermore, the populations that included the rheological data collected at the highest temperatures, close to denaturation, were more dispersed (e.g., run 6 vs. run 8, run 7 vs. run 9, or run 13 vs. run 14; see Table 4). Increasing the concentration exponentially brought the data closer to the null hypothesis; however, as shown in Table 3, there was a maximum concentration at which measurements could be made, as at 15% wt measurement was not possible.

All *p*-values for the machine learning adjusted data were larger, approaching the null hypothesis, which demonstrated the usefulness of applying the adjustments to increase the predictive performance of the method. Figure 5 shows how the good fit of the data, using the computer tool, enabled curves suitable for feeding the software to be obtained (i.e., curves that increased the cut-off frequency with temperature, as expected). However, as mentioned above, for concentrations of 15% by weight, a gel-like behavior was again obtained, which could not be adjusted. This was possibly due to the movement restrictions and the sensitivity of the entanglements to the chosen test conditions. Only a curve near 28 °C and out of the linear viscoelastic range could be obtained for this sample (see Figure 5e). The lack of data at other temperatures did not enable the master curve or the Mw distribution curve to be obtained at such high concentrations. All of these results demonstrated the sensitivity of the method to temperature and sample concentrations.

In light of the statistical tests and the data obtained, it can be said that the original collagen sample studied by the rheological method had a weight average Mw of 680 kDa, which was more than double what is usually reported in the literature for hydrolyzed collagen [1,3]. Furthermore, the method inferred that it was a polydisperse material (PDI ≈ 2). However, these data were homogeneous and reproducible only under very narrow conditions of concentration (i.e., approximately 4%; see Figure 6), temperature (i.e., high enough to induce molecular mobility, but far from T_d_), and deformation in the oscillatory tests (within the linear elastic range). Finally, as expected, the Gaussian curves depicted in Figure 6 show a non-uniform distribution of molecular weights due to the complexity of the system studied.

### 3.3. Validation of the Rheological Method

#### 3.3.1. Application of the Method to Samples of Different Mw

Although the proposed method was able to quantify high Mw molecules of collagen fibers, it was not possible to quantitatively verify whether the given numerical value was correct because, as described above, no traditional characterization method can do so (due to the need for sample digestion, high temperatures, low detection capacity, etc.).

To validate the rheological technique, the following methodology was designed. Three collagen samples were subjected to thermomechanical treatments at different times (see Section 2) to obtain samples of different Mw and MwD (based on the assumption that the longer the treatment time, the greater the fragmentation of the chains of the samples studied and therefore the lower the Mw and/or the widening of the molecular size distribution). Once the samples were obtained, they were diluted to 4% wt, according to the results obtained in the previous section.

The same test conditions were applied to these samples and they were found to be within the linear viscoelastic region of their untreated analogue. The mathematical fitting of the data was also applied. The Mw results obtained are shown in Table 5.

Homogeneous data were found for all samples when the Mw averages were plotted at a temperature of 28 °C; i.e., low dispersion and independence with temperature were observed. Moreover, the Mw values followed the expected trend: the longer the treatment time, the more polydisperse materials with shorter fibers were obtained (see Table 5 and Figure 7). This experiment qualitatively validated the usefulness of the method. However, it is important to note that between the original sample (4% wt) and the sample exposed to thermo–mechanical treatment for low periods (5 min), there was hardly any difference in the Mw curves. The method possibly lacked the sensitivity to capture all of the newly formed low Mw sections. There must be sufficient microstructural differences in the sample for the method to be quantitatively valid. ANOVAs for these populations reported *p*-values close to 0.05 in all cases, demonstrating the reproducibility of the method.

#### 3.3.2. Comparison with Traditional Measurement Techniques

Finally, the lowest molecular weight samples studied (4% wt_10 and 4% wt_15) were analyzed by SDS-PAGE and GPC, applying different sample digestion processes. The results are shown in Figure 8.

As shown in Figure 8, the electrophoretic pattern of both samples was very similar, showing a typical profile of type I collagen: a band with a molecular weight of around 120 kDa (α_1_); another band with a slightly lower molecular weight (α_2_) and the dimer (β) with a molecular weight of around 220 kDa [39,40]. In both cases, the insoluble material was observed at the top of the gel, which could correspond to high molecular weight structures with cross-links that prevented their dissolution in the electrophoresis buffer. This showed that the samples studied still had a high molecular weight and that this type of analysis could not be applied. This also shows the importance of the rheological technique that was studied.

Although no major differences were observed in the electrophoretic pattern of the samples that were previously lyophilized and those that were directly solubilized in the electrophoresis buffer, in the case of the latter there were signs of γ component (300 kDa to 400 kDa) and better solubilization of the collagen (i.e., less presence of aggregates that did not enter the gel). Even so, the method remains invalid. It should be noted that the Mw Gaussians obtained by the proposed rheological technique did not take into account the small chains detected in the electrophoresis pattern (i.e., the α and β chains of 120 kDa and 220 kDa, respectively). The smallest sizes falling within the MwD bell, as shown in Figure 7, were of the order of 500 kDa. Two conclusions can be drawn from this: either the samples obtained had very small amounts of low molecular weight, whose relaxations could not be detected in the oscillatory assays, or the method was not sufficiently sensitive for these populations, regardless of their content.

Finally, as shown in Figure 9, the chromatographic profile of the analyzed samples, both the 4% wt_15 sample (Figure 9a) and the 4% wt_10 sample (Figure 9b) were also very similar. In both samples, two peaks were observed that were more or less distinct. They corresponded to retention times of 45 and 48 min, to the β-dimer or component (calculated molecular weight: 220 kDa, polydispersity index: 1.03), and to the α-chains (calculated molecular weight: 112 kDa, polydispersity index: 1.07), respectively. As exhibited by the electrophoresis patterns, this showed that these low molecular weight chains were present in the samples studied.

In contrast to the electrophoresis pattern, the GPC results showed, in both samples, the presence of high molecular weights corresponding to retention times ranging from 35 to 44 min (calculated molecular weight: 400 kDa to 500 kDa, polydispersity index: 1.20), which fell within the prediction of the data obtained by rheology. Furthermore, a comparison of both GPC curves showed that for the 4% wt_15 sample, this high Mw population was shifted to higher retention times, i.e., lower Mw, which was to be expected because these chains underwent more scissions than did their analogue.

Finally, the application of the traditional GPC and SDS-PAGE techniques and the proposed rheology technique have their advantages and disadvantages. However, the rheological method has proved to be a very useful technique for comparing high molecular weight types of collagen and for opening up new possibilities of high value-added uses for their non-hydrolyzed or fragmented chains.

## 4. Conclusions

The results demonstrate that the rheological method is a valid technique for characterizing high-molecular-weight collagen fibers that cannot be identified by conventional characterization methods. The adjustments made to the obtained data from the frequency sweeps made it possible to obtain rheological curves that are easily translated into molecular-weight Gaussians curves, employing conventional software. It was shown that the proposed method is highly dependent on sample concentration and test conditions and lacks the sensitivity to identify low Mw sections. On the other hand, the proposed method is a simple method, does not require the use of solvents, and is easy to implement. It helps in readily distinguishing the microstructures of collagen gels and in finding the structure–properties relationship of these complex molecules so that more advanced biomaterials may be designed.

## Figures and Tables

**Figure 1 polymers-14-03683-f001:**
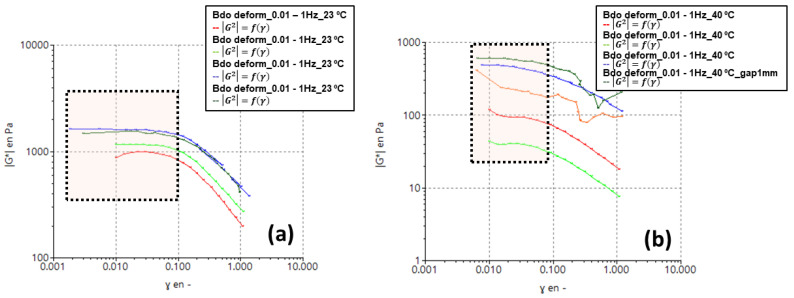
Strain sweeps at different frequencies to determine the linear viscoelastic region of a 2% wt sample (shaded area) at the initial and final test temperature: (**a**) 23 °C and (**b**) 40 °C.

**Figure 2 polymers-14-03683-f002:**
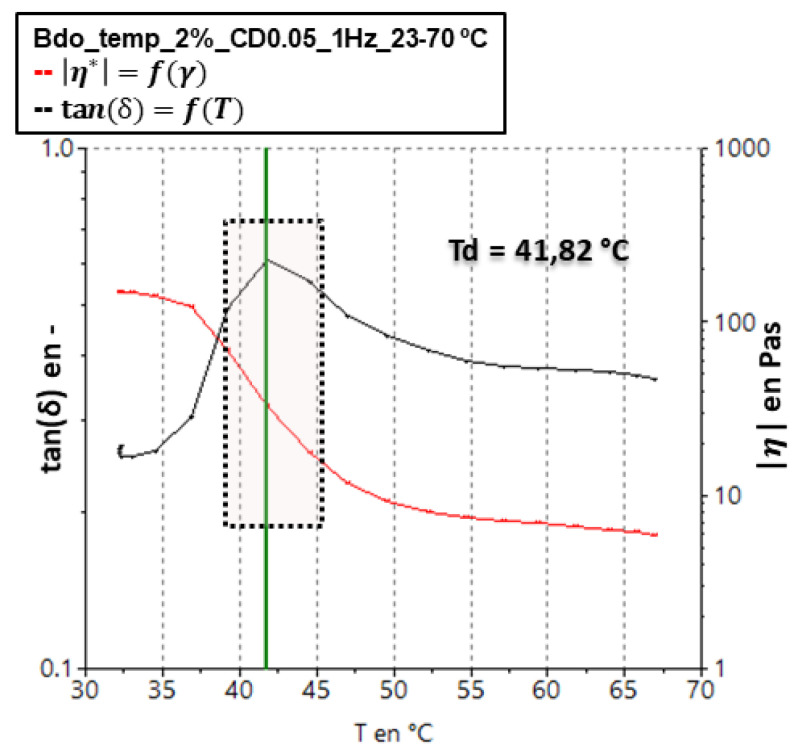
Determination of the denaturation temperature of the collagen mass at 2% wt.

**Figure 3 polymers-14-03683-f003:**
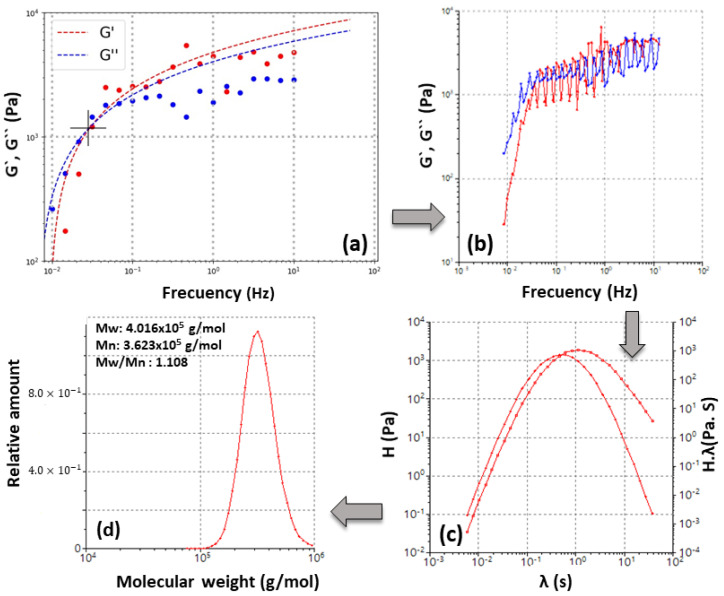
The sequence of steps to apply the proposed method to obtain the MwD by the rheological route: (**a**) fitting applied to the G′ and G″ curves; as can be observed, there are inhomogeneities in the experimental data when working near T_d;_ (**b**) obtaining the master curve at temperature defined with the TTS time–temperature superposition principle; (**c**) obtaining the relaxation spectrum; and (**d**) solving the equations and plotting the MwD of the sample.

**Figure 4 polymers-14-03683-f004:**
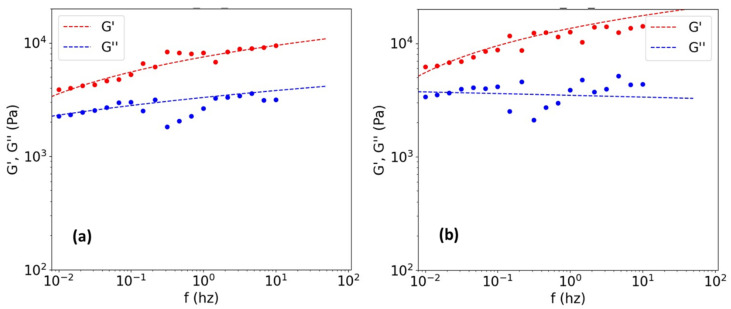
Gel-like behavior of 1% wt diluted samples at low temperatures of (**a**) 23 °C and (**b**) 28 °C.

**Figure 5 polymers-14-03683-f005:**
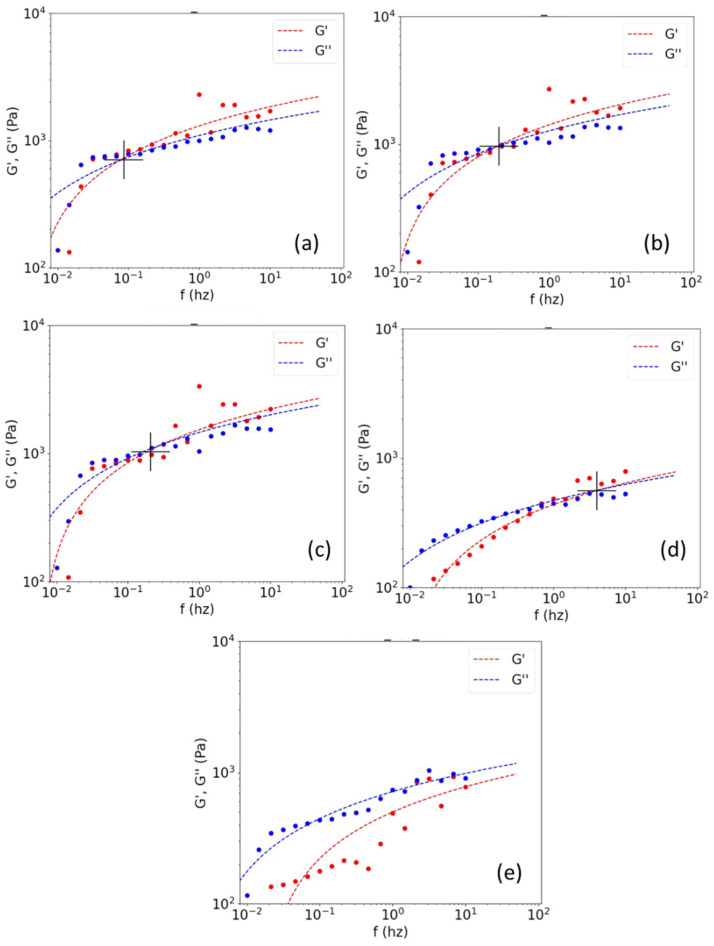
Frequency sweeps and their fittings within the viscoelastic range for the 5% wt sample at (**a**) 23 °C, (**b**) 28 °C, (**c**) 32 °C, and (**d**) 36 °C, and (**e**) for the 15% wt sample at 28 °C.

**Figure 6 polymers-14-03683-f006:**
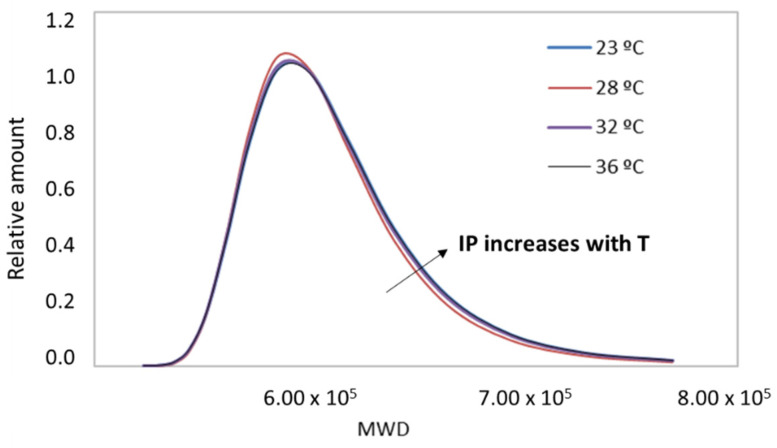
Distribution of molecular weights obtained with RheoWin software for the 4% wt sample at different test temperatures.

**Figure 7 polymers-14-03683-f007:**
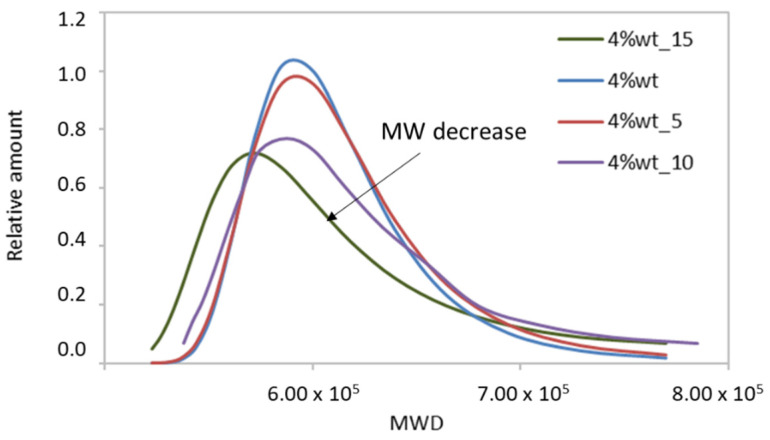
Distribution of molecular weights obtained with RheoWin software for 4% wt samples with different thermo–mechanical treatments.

**Figure 8 polymers-14-03683-f008:**
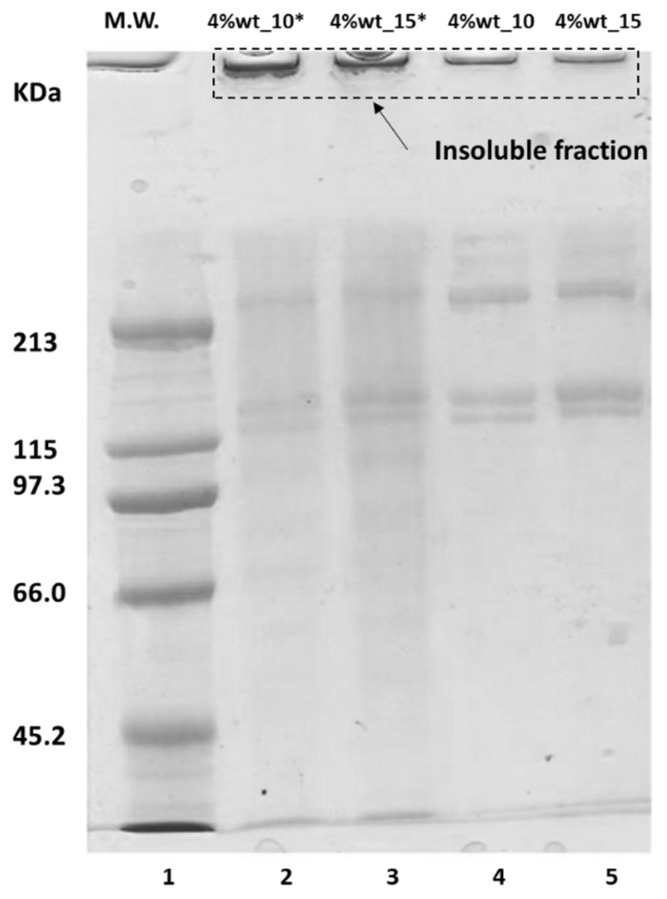
Electrophoresis pattern of the samples. 1: molecular weight standard; 2 (sample 4% wt_10*): 2 mg/mL (sample previously lyophilized and resuspended in 0.1 N acetic acid); 3 (sample 4% wt_15*): 2 mg/mL (sample previously lyophilized and resuspended in 0.1 N acetic acid); 4 (sample 4% wt_10): sample directly resuspended in sample buffer (29 mg/mL); and 5 (sample 4% wt_15): sample directly resuspended in sample buffer (29 mg/mL).

**Figure 9 polymers-14-03683-f009:**
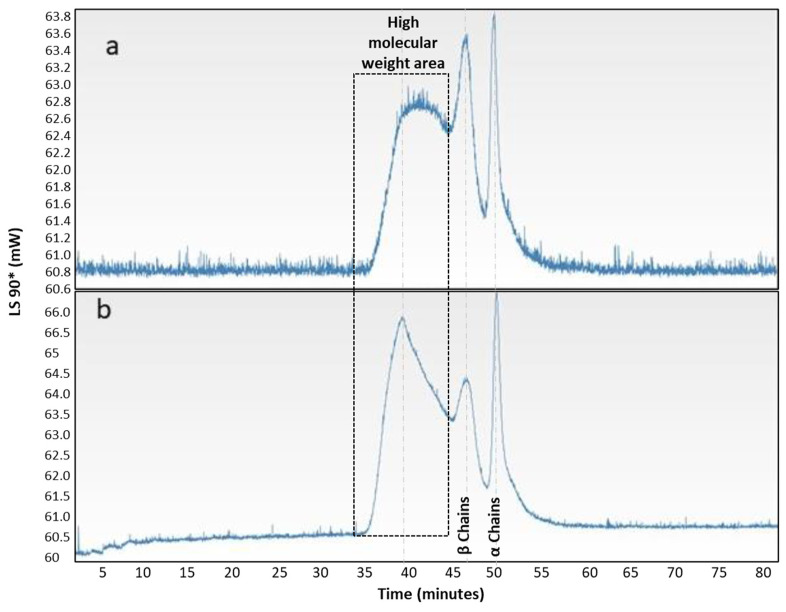
GPC curves for (**a**) 4% wt_15 y (**b**) 4% wt_10.

**Table 1 polymers-14-03683-t001:** Constant experimental test parameters used for collagen gels under stress or deformation modality.

	Experimental Parameters									
**Stress (Pa) (CS)** ^a^	10			50				100		
**Strain or Deformation (CD)** ^b^	0.01			0.05				0.5		
**Temperature (°C)**	23	25	28		32	34	36		38	40
**Gap (mm)**	1					2				

^a^ CS: constant stress; ^b^ CD: constant deformation

**Table 2 polymers-14-03683-t002:** Value of the denaturation temperatures measured for each concentration of the studied collagen masses.

Sample (% wt)	Denaturation Temperature ± 1.50 (°C)
1	41.94
2	40.82
3	38.41
4	38.15
5	38.12
15	37.20

**Table 3 polymers-14-03683-t003:** Molecular weights and their polydispersity index for samples with different concentrations.

**Sample ^b^**	**1% wt (Experimental)**				**1% wt (Theoretical)**			
**Exp. Cond. ^a^**	**CS: 10 Pa, Gap: 2 mm**							
**Temp. (°C)**	**23**	**28**	**34**	**40**	**23**	**28**	**34**	**40**
Mw (g/mol) ^d^	- ^c^	-	4.2 × 10^4^ ± 4.0 × 10^3^	-	1.7 × 10^5^ ± 1.0 × 10^5^	3.5 × 10^5^ ± 1.0 × 10^4^	5.7 × 10^5^ ± 1.0 × 10^4^	1369 ± 200
Mn (g/mol) ^e^	-	-	3.5 × 10^4^ ± 3.5 × 10^3^	-	1.4 × 10^5^ ± 1.5 × 10^5^	3.0 × 10^5^ ± 2.0 × 10^4^	4.2 × 10^4^ ± 1.0 × 10^4^	1166 ± 420
PDI ^f^	-	-	1.2 ± 0.7	-	1.2 ± 0.9	1.2 ± 0.6	1.4 ± 0.9	1.2 ± 0.9
**Sample**	**2% wt (Experimental)**				**2% wt (Theoretical)**			
**Exp. Cond.**	**CD: 0.5, Gap: 2 mm**							
**Temp. (°C)**	**23**	**28**	**34**	**40**	**23**	**28**	**34**	**40**
Mw (g/mol)	-	5.0 × 10^5^ ± 4.0 × 10^4^	5.5 × 10^5^ ± 2.0 × 10^5^	-	1.0 × 10^6^ ± 4.0 × 10^5^	5.5 × 10^5^ ± 2.0 × 10^5^	5.5 × 10^5^ ± 2.0 × 10^5^	-
Mn (g/mol)	-	3.1 × 10^5^ ± 2.0 × 10^4^	3.5 × 10^5^ ± 7.0 × 10^3^	-	5.8 × 10^5^ ± 3.0 × 10^2^	7.2 × 10^5^ ± 1.0 × 10^5^	3.5 × 10^5^ ± 7.0 × 10^3^	-
PDI	-	1.6 ± 0.8	1.6 ± 1.0	-	1.7 ± 0.7	1.9 ± 0.8	1.3 ± 1.1	-
**Sample**	**3% wt (Experimental)**				**3% wt (Theoretical)**			
**Exp. Cond.**	**CD: 1.0, Gap: 2 mm**							
**Temp. (°C)**	**23**	**28**	**32**	**36**	**23**	**28**	**32**	**36**
Mw (g/mol)	1.7 × 10^6^ ± 4.0 × 10^3^	7.3 × 10^5^ ± 4.0 × 10^3^	6.8 × 10^5^ ± 1.5 × 10^2^	5.2 × 10^5^ ± 0.5 × 10^2^	7.2 × 10^5^ ± 1.5 × 10^2^	8.5 × 10^5^ ± 1.5 × 10^2^	6.5 × 10^5^ ± 1.0 × 10^2^	4429 ± 120
Mn (g/mol)	9.4 × 10^5^ ± 3.0 × 10^3^	4.2 × 10^5^ ± 1.0 × 10^2^	3.7 × 10^5^ ± 3.0 × 10^2^	3.0 × 10^5^ ± 1.0 × 10^3^	3.7 × 10^5^ ± 0.5 × 10^2^	4.4 × 10^5^ ± 4.0 × 10^2^	3.4 × 10^5^ ± 3.5 × 10^2^	3163 ± 256
PDI	1.8 ± 0.2	1.7 ± 0.4	1.8 ± 0.3	1.7 ± 0.8	1.9 ± 0.1	1.9 ± 0.2	1.9 ± 0.4	1.4 ± 1.1
**Sample**	**4% wt (Experimental)**				**4% wt (Theoretical)**			
**Exp. Cond.**	**CS: 300 Pa, Gap: 0.5 mm**							
**Temp. (°C)**	**23**	**28**	**32**	**36**	**23**	**28**	**32**	**36**
Mw (g/mol)	5.2 × 10^5^ ± 1.0 × 10^2^	6.1 × 10^5^ ± 9.0 × 10^1^	6.9 × 10^5^ ± 1.5 × 10^2^	-	7.6 × 10^5^ ± 1.5 × 10^2^	6.7 × 10^5^ ± 1.0 × 10^3^	8.1 × 10^5^ ± 1.0 × 10^3^	-
Mn (g/mol)	2.4 × 10^5^ ± 2.5 × 10^2^	3.0 × 10^5^ ± 3.0 × 10^2^	3.6 × 10^5^ ± 1.0 × 10^3^	-	3.8 × 10^5^ ± 1.5 × 10^3^	3.7 × 10^5^ ± 2.5 × 10^2^	4.3 × 10^5^ ± 1.0 × 10^3^	-
PDI	2.1 ± 0.5	2.0 ± 0.2	1.9 ± 0.4	-	2.0 ± 0.1	1.8 ± 0.4	1.9 ± 0.3	-
**Sample**	**5% wt (Experimental)**				**5% wt (Theoretical)**			
**Exp. Cond.**	**CS: 500 Pa, Gap: 1 mm**							
**Temp. (°C)**	**23**	**28**	**34**	**36**	**23**	**28**	**34**	**36**
Mw (g/mol)	6.3 × 10^5^ ± 1.5 × 10^2^	6.8 × 10^5^ ± 3.0 × 10^2^	7.1 × 10^5^ ± 1.0 × 10^2^	-	7.6 × 10^5^ ± 1.0 × 10^2^	6.9 × 10^5^ ± 0.9 × 10^2^	7.2 × 10^5^ ± 1.5 × 10^2^	-
Mn (g/mol)	3.5 × 10^5^ ± 1.0 × 10^3^	3.6 × 10^5^ ± 1.5 × 10^3^	3.7 × 10^5^ ± 2.0 × 10^3^	-	3.6 × 10^5^ ± 1.0 × 10^3^	3.1 × 10^5^ ± 5.0 × 10^2^	3.6 × 10^5^ ± 3.5 × 10^3^	-
PDI	1.8 ± 0.5	1.9 ± 0.5	1.9 ± 0.2	-	2.1 ± 0.2	2.2 ± 0.5	2.0 ± 0.3	-
**Sample**	**15% wt (Experimental)**				**15% wt (Theoretical)**			
**Exp. Cond.**	**CD: 0.5, Gap: 2 mm**							
**Temp. (°C)**	**23**	**28**	**34**	**36**	**23**	**28**	**34**	**36**
Mw (g/mol)	-	-	-	-	-	-	-	-
Mn (g/mol)	-	-	-	-	-	-	-	-
PDI	-	-	-	-	-	-	-	-

^a^ The best experimental conditions for performing the oscillatory tests are shown in the boxes on the top for each concentration. Shaded data denote data collected outside the linear viscoelastic range. In the case of the 15% wt sample, rheological curves were obtained but it was not possible to translate them into Gaussian Mw. ^b^ Experimental data means raw data and theoretical data means data fitted with the computer tool. ^c^ Boxes filled with—symbol means that it was not possible to obtain data; gel-like behavior was obtained outside the linear viscoelastic region or data were obtained very close to the denaturation of the material. ^d^ Weight average molecular weight. ^e^ Number average molecular weight. ^f^ Polydispersity index.

**Table 4 polymers-14-03683-t004:** *p*-value from analysis of variance for each population under study.

Run	Studied Population			*p*-Value
	Concentration (% wt)	Temperature (°C)	Data	
**1**	1	34	Experimental	-
**2**	1	23, 28, 34, 40	Theoretical	5 × 10^−18^
**3**	1	23, 28, 34	Theoretical	7 × 10^−12^
**4**	2	28, 34	Experimental	3 × 10^−17^
**5**	2	23, 28, 34	Theoretical	2 × 10^−10^
**6**	3	23, 28, 32, 36	Experimental	6 × 10^−13^
**7**	3	23, 28, 32, 36	Theoretical	2 × 10^−9^
**8**	3	23, 28, 32	Experimental	4 × 10^−8^
**9**	3	23, 28, 32	Theoretical	8 × 10^−6^
**10**	4	23, 28, 32	Experimental	4 × 10^−6^
**11**	4	23, 28, 32	Theoretical	1 × 10^−5^
**12**	5	23, 28, 32	Experimental	2 × 10^−4^
**13**	5	23, 28, 32	Theoretical	7 × 10^−3^
**14**	5	23, 28	Theoretical	1 × 10^−3^
**15**	3, 4, 5	23	Theoretical	5 × 10^−5^
**16**	3, 4, 5	28	Theoretical	3 × 10^−3^
**17**	3, 4, 5	32	Theoretical	9 × 10^−5^
**18**	4, 5	28	Theoretical	0.02

**Table 5 polymers-14-03683-t005:** Molecular weights and their polydispersity index for 4% wt samples with different thermo-mechanical treatments.

Exp. Cond.	CS: 300 Pa, Gap: 0.5 mm							
Temp. (°C)	23	28	32	36	23	28	32	36
**Sample**	**4% wt (Theoretical)**				**4% wt*_*5 (Theoretical)**			
Mw (g/mol)	7.6 × 10^5^ ± 1.5 × 10^2^	6.7 × 10^5^ ± 1.0 × 10^3^	8.1 × 10^5^ ± 1.0 × 10^3^	-	6.2 × 10^5^ ± 1.0 × 10^3^	6.9 × 10^5^ ± 3.0 × 10^2^	6.5 × 10^5^ ± 3.5 × 10^3^	-
Mn (g/mol)	3.8 × 10^5^ ± 1.5 × 10^3^	3.7 × 10^5^ ± 2.5 × 10^2^	4.3 × 10^5^ ± 1.0 × 10^3^	-	2.8 × 10^5^ ± 1.0 × 10^4^	2.4 × 10^5^ ± 6.5 × 10^2^	2.8 × 10^5^ ± 3.5 × 10^3^	-
PDI	2.0 ± 0.1	1.8 ± 0.4	1.9 ± 0.3	-	2.2 ± 0.1	2.8 ± 0.6	2.3 ± 0.5	-
**Sample**	**4% wt*_*10 (Theoretical)**				**5% wt*_*15 (Theoretical)**			
Mw (g/mol)	5.8 × 10^5^ ± 1.5 × 10^2^	6.1 × 10^5^ ± 3.0 × 10^2^	5.2 × 10^5^ ± 1.0 × 10^2^	-	5.5 × 10^5^ ± 1.0 × 10^2^	5.4 × 10^5^ ± 0.9 × 10^2^	5.7 × 10^5^ ± 1.5 × 10^2^	-
Mn (g/mol)	2.1 × 10^5^ ± 3.0 × 10^3^	2.0 × 10^5^ ± 4.5 × 10^3^	1.6 × 10^5^ ± 1.0 × 10^4^	-	1.6 × 10^5^ ± 1.0 × 10^4^	1.7 × 10^5^ ± 2.0 × 10^3^	1.9 × 10^5^ ± 4.5 × 10^3^	-
PDI	2.8 ± 0.4	3.1 ± 0.3	3.2 ± 0.5	-	3.4 ± 0.6	3.2 ± 0.1	3.0 ± 0.3	-

## Data Availability

Not applicable.

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
