# Peer review of "Rheological Method for Determining the Molecular Weight of Collagen Gels by Using a Machine Learning Technique"

_polymers, 2022, doi:10.3390/polym14173683_

Round 1
Reviewer 1 Report
The article "Rheological method for determining the molecular weight (MW) of collagens gels by using a machine learning technique" describes the implementation of mathematical models toward determination of molecular weight for a biomolecule (namely collagen), using rheological data. It is a valuable study that can be published after authors address the following problems:
The English language needs some minor polishing for style and typos.
Please improve the quality of images in Figure 1 and 2 and especially improve the legend readability.
In introduction a stronger recent literature survey is necessary, especially on previous literature reports on the collagen applications. I suggest that the authors find an opportunity to mention the following publications in Introduction section, in order to broaden the range of collagen uses: doi: 10.3390/Polym14122430; doi: 10.33263/Briac114.1198511995
Abbreviations must be explained at first use (GPC appears at row 107 and explanation is at row 267 after other two encounters). SDS-PAGE abbreviation comes from sodium dodecyl sulphate–polyacrylamide gel electrophoresis, not just from colloquially electrophoresis, just as MALDI-TOF is not just mass spectrometry. There are other abbreviations not explained like PTFE
Use uniform notation for measurement units (now for litre are used both l and L like at rows 275 or 278 and rows 259, 261 or 277 but also elsewhere across the manuscript, like in figure 8). Personally, I would recommend the use of L.
The values presented in Table 3 have some problems that must be addressed: either typos or if they are corrected then authors should try explain and interpret these data (as this means that data isn't normally distributed, with a high degree of uncertainty and authors must explain it, and also argument the use of average and SD instead of median and range):
First data row, experimental 34 column: 4.2x104± 4.0x104 – standard deviation is almost equal with the average value.
First and second data row, theoretical 40 column: 1.166±4.2 – a SD almost four times greater than the average value.
In Figure 8 please differentiate between samples 2-4 and 3-5 also in the image legend not only in caption.
In the rows before Figure 8 (375-376) authors state that “Finally, the lowest molecular weight samples studied (5% wt_10 and 5% wt_15) were analysed by SDS-PAGE and GPC. The results are shown in figure 8”. However, in Figure 8 samples are labeled 4% wt_10 and 4% wt_15. Please correct it.
Author Response
Point 1: The English language needs some minor polishing for style and typos.
Response 1: The recommendation has been taken into account. The manuscript has been thoroughly revised; in addition, many paragraphs have been rewritten for better understanding, as also suggested by other reviewers.
Point 2: Please improve the quality of images in Figure 1 and 2 and especially improve the legend readability.
Response 2: Both figures have been improved.
Point 3: In introduction a stronger recent literature survey is necessary, especially on previous literature reports on the collagen applications. I suggest that the authors find an opportunity to mention the following publications in Introduction section, in order to broaden the range of collagen uses: doi: 10.3390/Polym14122430; doi: 10.33263/Briac114.1198511995
Response 3: We are grateful for the recommendation made. After reading the suggested papers, they have been added as part of the introduction to complement the information about the current use of collagens. For traceability, the changes have been highlighted in yellow.
Point 4: Abbreviations must be explained at first use (GPC appears at row 107 and explanation is at row 267 after other two encounters). SDS-PAGE abbreviation comes from sodium dodecyl sulphate–polyacrylamide gel electrophoresis, not just from colloquially electrophoresis, just as MALDI-TOF is not just mass spectrometry. There are other abbreviations not explained like PTFE
Response 4: All abbreviations have been explained correctly and in order of appearance in the text.
For traceability, the changes have been highlighted in yellow.
Point 5: Use uniform notation for measurement units (now for litre are used both l and L like at rows 275 or 278 and rows 259, 261 or 277 but also elsewhere across the manuscript, like in figure 8). Personally, I would recommend the use of L.
Response 5: The abbreviation for litres has been uniformly stated as L, throughout the manuscript. For traceability, the changes have been highlighted in yellow.
Point 6: The values presented in Table 3 have some problems that must be addressed: either typos or if they are corrected then authors should try explain and interpret these data (as this means that data isn't normally distributed, with a high degree of uncertainty and authors must explain it, and also argument the use of average and SD instead of median and range):
First data row, experimental 34 column: 4.2x104± 4.0x104 – standard deviation is almost equal with the average value.
First and second data row, theoretical 40 column: 1.166±4.2 – a SD almost four times greater than the average value.
Response 6: We are grateful for the important correction suggested. The data have been thoroughly checked and explained in the manuscript. In addition, An analysis of variance (ANOVA) was included in order to reach a better conclusion on the statistical validity of the data. For traceability, the changes have been highlighted in yellow.
Point 7: In Figure 8 please differentiate between samples 2-4 and 3-5 also in the image legend not only in caption.
In the rows before Figure 8 (375-376) authors state that “Finally, the lowest molecular weight samples studied (5% wt_10 and 5% wt_15) were analysed by SDS-PAGE and GPC. The results are shown in figure 8”. However, in Figure 8 samples are labeled 4% wt_10 and 4% wt_15. Please correct it.
Response 7: Figure 8 has been modified. Asterisks have been added to lanes 2 and 3 to show that these are different samples from those shown in lanes 4 and 5. The figure legend explains the different treatments applied.
The sample analysed in figure 8 corresponds to 4% wt and not to 5% wt. The error has been corrected.
The data have been thoroughly checked and explained in the manuscript. For traceability, the changes have been highlighted in yellow.
We thank you for all your valuable contributions and remain at your disposal for any further questions or suggestions.
Yours sincerely,

Reviewer 2 Report
Interesting results and novelty work. A paper focuses on Rheological method for determining the molecular weight (MW) of collagens gels by using a machine learning technique. Though the intention of the authors is highly commendable, there is lot of problems particularly in the presentation throughout the manuscript. Besides there are many grammatical mistakes throughout the manuscript, particularly in respect of use of singular and plural with the subject or verb. In view of the above comments, whole manuscript should be properly written to make it acceptable by polymers journal. I highly recommended this article to be accepted and published in the revised version.
Abstract:
The abstract given here starts without any background for the present work. Of course, it contains brief details about experimental aspects and the obtained results. However this abstract does not follow the norm of an abstract, which should state briefly:
1. The purpose of the study undertaken, what are you trying to solve
2. brief mention of experimental aspects (without using abbreviations)
3. highlights of the results numerically
4. Important conclusions based on the obtained results
5. Potential applications
Therefore, it is suggested that the Abstract to be modified as per the suggestions given above.
Introduction
Introduction section is long with a many references based on the literature survey conducted by the authors. This is very good. However, this lacks in proper presentation of literature survey, which should have been systematic whereby existing scientific gaps should have been brought out. This should have given justification for the present study, which should be followed by the objectives of this study. In fact there is large amount of literature available on the characterization of molecular weight (MW) using a machine learning technique. Similarly, a large number of methods to obtain these materials have been used mentioning their advantages and limitations. None of these have been brought out in this study whereby the authors have not justified why they have chosen the method they have used in their study. It should be noted that normally 'Introduction' should give brief background through literature survey for the study citing previous published work where-by scientific gaps that exist should be brought out. This would have led to justification for the present study. It is therefore suggested that ‘Introduction Section’ should be revised as suggested above because this Section is an important one from the point of view of taking up the present study.
In my opinion the paper will have good merit if such applications can be demonstrated and reported. Can you give some example?
Use at least one sentence to describe each cited paper in the references; avoid linking several papers and describe it.
….industries [2,3], cosmetics [4], food [5–7], packaging [8];…
Traditional molecular weight measurement techniques represent destructive methods that fragment the protein and do not give a true value of the molecular weight distribution. Where do you get this information? Please do cite.
. To achieve this goal, it has been necessary to use artificial intelligence techniques to fit, and predict, the theoretical data. In recent years these techniques have shown their potential in the identification and characterisation of biomolecules [28–30] and will be used for the first time in the characterisation of high molecular weight collagens.
Last sentence should be your objective and method to achieve it. Please revise.
Materials and Methods:
Normally, this section should have two main subsections. The first one is Materials which should give details of all materials used in the study, where from they were procured, known characteristics, if available (for e.g. bovine, collagen etc, where do you get it, what is the purity of the chemical and etc.).
Diluted samples of collagen were then prepared in acetic acid at concentrations of 1, 2, 3, 4, 5 and 15% w/w… until last sentence. These sentences should not be in Section 2.1. Please remove it. Only report what material you used, purity, where do you purchased/obtained.
The second subsection should be Methods, where methodologies used in the study should be given in a systematic way using sub section with numbers for each of the properties. First the processing or preparation aspects of the final material should be given followed by the characterization of prepared materials including preparation of samples for any specific property or morphology studies should be presented in a systematic way. Here one should also clearly mention the number of samples used, any standards followed for variety of properties, make and model of the instruments used for characterization, their accuracy and experimental conditions used, etc.
It should be known to the authors when one publishes any scientific paper, the results presented therein should be such they should be reproducible by any other person when the experiment is repeated using the same materials. In the present paper, it would be difficult for any other person to repeat the experiments because the chosen materials do not have any pre-history, which is required for other researchers to carryout experiments to check the possible reproducibility of the procedure adopted by these authors.
Some of the paragraph should be under results and discussion and if it is already there then this becomes repetition and hence can be deleted. Secondly, this Section is methods and hence only results should be mentioned and then it should be discussed preferably comparing it with earlier reported similar results by other researchers.
Formula that you generate come from where? Any ref?
Figure 1 and 2 is blur. Please revise.
Table 2… wt% should delete… only insert next to Sample (% wt.) …. No need to repeat many time.
Results & Discussion
Well written and easy for the reader to understand what the authors have conveyed.
Some of the paragraph should be under Methods and if it is already there then this becomes repetition and hence can be deleted. Secondly, this Section is Results & Discussion and hence only results should be mentioned and then it should be discussed preferably comparing it with earlier reported similar results by other researchers.
Throughout the manuscript, there are no comparison had been done with other published journal. Therefore, please support your statements with other researcher’s work in the section result and discussion. It should be discussed preferably comparing it with earlier reported similar results by other researchers.
Please write your sentence clearly. This is some tips.
In order to make paper easily understandable, for every sentence you write you ask yourself these two questions:
(question 1) why?
(question 2) where is the evidence and is the evidence solid evidence or very weak evidence? This means if you cannot answer these two questions, the sentence is no longer easy to read by the readers.
for example: "a woman walks quickly to the superstore and wears blue color coats"
question 1: why women why not a man? why walk so quickly? why go to the superstore? why wear blue color coats. Was that day a rainy days or about to rain and that is why the person walk so quickly to avoid big rain forthcoming?
question 2: where is the evidence? by somebody sees her from far away or near? by close circuit television? or by a photographer taking the photo from far away ? does the photo show her face? is that blue coats a rainy coat?
How many sample did for each experiment? Please do ANNOVA test and standard deviation for all data collected and presented.
Conclusions
Conclusions given here are do not reflect what had been achieved including many speculations. It is too long and should be in 1 paragraph. Hence these need to be suitably modified. It may be remembered that this Section forms a summary of all the major observations/ results obtained. Accordingly, here presentation should consist of the main Results or the observations of the study in short sentences probably with bullet points. This should stand alone or form a subsection of a Discussion or Results Section. Hence better to rewrite this Section based on the comments given in the whole text.
General Comments:
The paper though contains some interesting results and novelty work, it lacks in its proper presentation in the whole manuscript. Of course there is need for better language throughout the manuscript. It is suggested that the authors should take the help of native English speaking person to take care of this problem. In view of these, the paper is highly recommended and should be accepted for publication in the revised form. It is suggested that the authors should revise the paper in the light of above comments/suggestions.
Author Response
Point 1: The abstract given here starts without any background for the present work. Of course, it contains brief details about experimental aspects and the obtained results. However, this abstract does not follow the norm of an abstract, which should state briefly:
- The purpose of the study undertaken, what are you trying to solve
- brief mention of experimental aspects (without using abbreviations)
- highlights of the results numerically
- Important conclusions based on the obtained results
- Potential applications
Therefore, it is suggested that the Abstract to be modified as per the suggestions given above.
Response 1: We welcome suggestions, the abstract has been revised and rewritten taking into account the recommendations.
Point 2: Introduction section is long with many references based on the literature survey conducted by the authors. This is very good. However, this lacks in proper presentation of literature survey, which should have been systematic whereby existing scientific gaps should have been brought out. This should have given justification for the present study, which should be followed by the objectives of this study. In fact there is large amount of literature available on the characterization of molecular weight (MW) using a machine learning technique. Similarly, a large number of methods to obtain these materials have been used mentioning their advantages and limitations. None of these have been brought out in this study whereby the authors have not justified why they have chosen the method they have used in their study. It should be noted that normally 'Introduction' should give brief background through literature survey for the study citing previous published work where-by scientific gaps that exist should be brought out. This would have led to justification for the present study. It is therefore suggested that ‘Introduction Section’ should be revised as suggested above because this Section is an important one from the point of view of taking up the present study.
In my opinion the paper will have good merit if such applications can be demonstrated and reported. Can you give some example?
Use at least one sentence to describe each cited paper in the references; avoid linking several papers and describe it.
….industries [2,3], cosmetics [4], food [5–7], packaging [8];…
Traditional molecular weight measurement techniques represent destructive methods that fragment the protein and do not give a true value of the molecular weight distribution. Where do you get this information? Please do cite.
To achieve this goal, it has been necessary to use artificial intelligence techniques to fit, and predict, the theoretical data. In recent years these techniques have shown their potential in the identification and characterisation of biomolecules [28–30] and will be used for the first time in the characterisation of high molecular weight collagens.
Last sentence should be your objective and method to achieve it. Please revise.
Response 2: The main thread of the introduction is based on demonstrating that the current techniques for characterising the molecular weights of macromolecules are not useful for measuring long collagen fibres. Therefore, the principle and limitation of each technique for this purpose is explained; we believe that this information will allow the reader to understand the problem and the importance of the solution presented. In response to the reviewer's suggestion, a more concise and focused introduction is presented in the new manuscript.
Point 3: Materials and Methods. Normally, this section should have two main subsections. The first one is Materials which should give details of all materials used in the study, where from they were procured, known characteristics, if available (for e.g. bovine, collagen etc, where do you get it, what is the purity of the chemical and etc.).
Diluted samples of collagen were then prepared in acetic acid at concentrations of 1, 2, 3, 4, 5 and 15% w/w… until last sentence. These sentences should not be in Section 2.1. Please remove it. Only report what material you used, purity, where do you purchased/obtained.
The second subsection should be Methods, where methodologies used in the study should be given in a systematic way using sub section with numbers for each of the properties. First the processing or preparation aspects of the final material should be given followed by the characterization of prepared materials including preparation of samples for any specific property or morphology studies should be presented in a systematic way. Here one should also clearly mention the number of samples used, any standards followed for variety of properties, make and model of the instruments used for characterization, their accuracy and experimental conditions used, etc.
It should be known to the authors when one publishes any scientific paper, the results presented therein should be such they should be reproducible by any other person when the experiment is repeated using the same materials. In the present paper, it would be difficult for any other person to repeat the experiments because the chosen materials do not have any pre-history, which is required for other researchers to carryout experiments to check the possible reproducibility of the procedure adopted by these authors.
Some of the paragraph should be under results and discussion and if it is already there then this becomes repetition and hence can be deleted. Secondly, this Section is methods and hence only results should be mentioned and then it should be discussed preferably comparing it with earlier reported similar results by other researchers.
Formula that you generate come from where? Any ref?
Figure 1 and 2 is blur. Please revise.
Table 2… wt% should delete… only insert next to Sample (% wt.) …. No need to repeat many time.
Response 3: The entire materials and methods section has been rewritten and organised with appropriate headings. All equations used have been better referenced. On the other hand, the quality of figures 1 and 2 has been improved.
Point 4: Results & Discussion
Well written and easy for the reader to understand what the authors have conveyed.
Some of the paragraph should be under Methods and if it is already there then this becomes repetition and hence can be deleted. Secondly, this Section is Results & Discussion and hence only results should be mentioned and then it should be discussed preferably comparing it with earlier reported similar results by other researchers.
Throughout the manuscript, there are no comparison had been done with other published journal. Therefore, please support your statements with other researcher’s work in the section result and discussion. It should be discussed preferably comparing it with earlier reported similar results by other researchers.
Please write your sentence clearly. This is some tips.
In order to make paper easily understandable, for every sentence you write you ask yourself these two questions:
(question 1) why?
(question 2) where is the evidence and is the evidence solid evidence or very weak evidence? This means if you cannot answer these two questions, the sentence is no longer easy to read by the readers.
for example: "a woman walks quickly to the superstore and wears blue color coats"
question 1: why women why not a man? why walk so quickly? why go to the superstore? why wear blue color coats. Was that day a rainy days or about to rain and that is why the person walk so quickly to avoid big rain forthcoming?
question 2: where is the evidence? by somebody sees her from far away or near? by close circuit television? or by a photographer taking the photo from far away? does the photo show her face? is that blue coats a rainy coat?
How many samples did for each experiment? Please do ANNOVA test and standard deviation for all data collected and presented.
Response 4: The discussion of results has been revised following the suggested guidelines. In addition, An analysis of variance (ANOVA) was included in order to reach a better conclusion on the statistical validity of the data. For traceability, the changes have been highlighted in yellow.
Point 5: Conclusions given here are do not reflect what had been achieved including many speculations. It is too long and should be in 1 paragraph. Hence these need to be suitably modified. It may be remembered that this Section forms a summary of all the major observations/ results obtained. Accordingly, here presentation should consist of the main Results or the observations of the study in short sentences probably with bullet points. This should stand alone or form a subsection of a Discussion or Results Section. Hence better to rewrite this Section based on the comments given in the whole text.
Response 5: In accordance with the suggested recommendations, the conclusion section has been rewritten in the new version of the manuscript.
Point 6: General Comments:
The paper though contains some interesting results and novelty work, it lacks in its proper presentation in the whole manuscript. Of course, there is need for better language throughout the manuscript. It is suggested that the authors should take the help of native English-speaking person to take care of this problem. In view of these, the paper is highly recommended and should be accepted for publication in the revised form. It is suggested that the authors should revise the paper in the light of above comments/suggestions.
We thank you for all your valuable contributions and remain at your disposal for any further questions or suggestions.
Yours sincerely,

Reviewer 3 Report
This paper dealt with Rheological method for determining the molecular weight (MW) of collagens gels by using a machine learning technique. There are some questions needed to be addressed.
Abstract: Q: I recommend rewrite Abstract section. The abstract should be clear and include purpose, methods, results and conclusion.
Materials and methods: Q: I recommend reorganize and revise this section and divide this section into subsections using appropriate subheadings. Cite the references for the methods used.
Lines 3, 132, 147, 215, 285, 342, 343, and 374. Q: Please delete the redundant punctuation marks.
Line 136: ~in the linear and no lineal viscoelastic region~ Q: Can you explain what the linear and no lineal viscoelastic region is? What does “no lineal” mean?
Author Response
Point 1: Abstract: Q: I recommend rewrite Abstract section. The abstract should be clear and include purpose, methods, results and conclusion.
Response 1: The recommendation has been taken into account. In addition to the abstract, the manuscript has been thoroughly revised, as also suggested by other reviewers.
Point 2: Materials and methods: Q: I recommend reorganize and revise this section and divide this section into subsections using appropriate subheadings. Cite the references for the methods used.
Response 2: The entire materials and methods section has been rewritten and organised with appropriate headings. All equations used have been better referenced.
Point 3: Lines 3, 132, 147, 215, 285, 342, 343, and 374. Q: Please delete the redundant punctuation marks.
Response 3: Excess punctuation marks have been corrected.
Point 4: Line 136: ~in the linear and no lineal viscoelastic region~ Q: Can you explain what the linear and no lineal viscoelastic region is? What does “no lineal” mean?
Response 4: As the reviewer rightly points out, the first time the concept of linear and non-linear viscoelastic region is mentioned, it is not defined. However, it is later clarified that "The linear viscoelastic region is denoted when the values of the modulus (G*) are independent of stress or strain".
For the better understanding of the reader the definition has been moved, where it is first mentioned.
Indeed, there is an error in the word "lineal", it should be "linear". This error has been corrected. For traceability, the changes have been highlighted in yellow.
We thank you for all your valuable contributions and remain at your disposal for any further questions or suggestions.
Yours sincerely,

Reviewer 4 Report
The article is related to the rheological method for determining the molecular weight (MW) of collagen gels by using a machine learning technique. The following significant specific comments are suggested to the authors.
1. The MW should be deleted from the title of this manuscript. And the MW should be written as Mw in the whole manuscript.
2. The authors have written long sentences several times in the manuscript's text. These sentences should be divided into short sentences.
3. In the Introduction part, several sentences are not supported by the references. It should be cited by the references. In line 31, although word should be deleted.
4. The authors wrote some sentences like Current work, such as that presented by, and Other authors such as; this type sentences should be rephrased in the whole manuscript.
5. In lines 72-79, the authors have described about the SEC techniques. What is the aim of describing this type of thing in the introduction? The reviewer has observed that the authors have described some unnecessary sentences in the introduction section. That sentence description does not match the title of this research. Thus, the text of the introduction should be revised again carefully.
6. Lines 106-111 and 155-156 should be rephrased. And kindly recheck lines 124-125. In line 149, what is m(W)? The resolution of figure 1, 2, and 3 are low, and the text of these figures are not readable.
The authors have written the abbreviation Td of denaturation temperature but sometimes as Td. It should be correct in the whole manuscript. Finally, the typo-error and grammatical errors should be carefully revised before publication in the Journal. Also, the English language deserves deep revision and improvement.
Author Response
The article is related to the rheological method for determining the molecular weight (MW) of collagen gels by using a machine learning technique. The following significant specific comments are suggested to the authors.
Point 1: The MW should be deleted from the title of this manuscript. and the MW should be written as Mw in the whole manuscript.
Response 1: All suggestions have been addressed. For traceability, the changes have been highlighted in yellow.
Point 2: The authors have written long sentences several times in the manuscript's text. These sentences should be divided into short sentences.
Response 2: The recommendation has been taken into account. The manuscript has been thoroughly revised; in addition, many paragraphs have been rewritten for better understanding, as also suggested by other reviewers.
Point 3: In the Introduction part, several sentences are not supported by the references. It should be cited by the references. In line 31, although word should be deleted.
Response 3: The introduction has been rewritten and special attention has been paid to the proper use of quotations.
Point 4: The authors wrote some sentences like Current work, such as that presented by, and Other authors such as; this type sentences should be rephrased in the whole manuscript.
Response 4: This recommendation has been addressed in the new introduction to the manuscript.
Point 5: In lines 72-79, the authors have described about the SEC techniques. What is the aim of describing this type of thing in the introduction? The reviewer has observed that the authors have described some unnecessary sentences in the introduction section. That sentence description does not match the title of this research. Thus, the text of the introduction should be revised again carefully.
Response 5: The main thread of the introduction is based on demonstrating that the current techniques for characterising the molecular weights of macromolecules are not useful for measuring collagen fibres. Therefore, the principle and limitation of each technique for this purpose is explained; we believe that this information will allow the reader to understand the problem and the importance of the solution presented. In response to the reviewer's suggestion, a more concise and focused introduction is presented in this document.
Point 6: Lines 106-111 and 155-156 should be rephrased. And kindly recheck lines 124-125. In line 149, what is m(W)? The resolution of figure 1, 2, and 3 are low, and the text of these figures are not readable.
Response 6: The lines mentioned have been rewritten, as suggested by the reviewer.
As for the definition of m(W), it is certainly confusing and is not explained in the text; it is a function of molecular motions as a function of sample weight. This has been clarified in the new version of the manuscript. As for the figures, they have been improved for better understanding.
Point 7: The authors have written the abbreviation Td of denaturation temperature but sometimes as Td. It should be correct in the whole manuscript. Finally, the typo-error and grammatical errors should be carefully revised before publication in the Journal. Also, the English language deserves deep revision and improvement.
Response 7: The recommendation has been taken into account. The manuscript has been thoroughly revised; in addition, many paragraphs have been rewritten for better understanding, as also suggested by other reviewers.
The nomenclature Td was standardised as Td. For traceability, the changes have been highlighted in yellow.
We thank you for all your valuable contributions and remain at your disposal for any further questions or suggestions.
Yours sincerely,

Round 2
Reviewer 1 Report
The authors have responded to my comments and have addressed all my concerns, substantially improving the manuscript, therefore, I suggest publishing the paper in the current form. Nevertheless, the authors need to improve the quality of the figures (v2 figures are worst than v1 original submission).
Author Response
Pont 1: The authors have responded to my comments and have addressed all my concerns, substantially improving the manuscript, therefore, I suggest publishing the paper in the current form. Nevertheless, the authors need to improve the quality of the figures (v2 figures are worst than v1 original submission).
Response 1: Figures 1, 2 and 3 have been modified and appended to the new manuscript in a new format so as not to lose resolution.
We thank you for all your valuable contributions and remain at your disposal for any further questions or suggestions.
Yours sincerely,

Reviewer 3 Report
The authors have addressed all of my concerns.
Author Response
Pont 1: The authors have addressed all of my concerns.
Response 1: -
We thank you for all your valuable contributions and remain at your disposal for any further questions or suggestions.
Yours sincerely,

Reviewer 4 Report
Accept
Author Response
Pont 1: Accept
Response 1: -
We thank you for all your valuable contributions and remain at your disposal for any further questions or suggestions.
Yours sincerely,
